# Enabling Real-Time Inference in Online Continual Learning via Device-Cloud Collaboration

## Abstract

Online continual learning (CL) is becoming a mainstream paradigm to learn incrementally from task streams without forgetting previously learned knowledge. However, current online CL primarily focuses on the learning performance, such as avoiding catastrophic forgetting, neglecting the critical demands of real-time inference. As a result, the performance of real-time inference in online CL degrades significantly due to frequent data distribution variations and time-consuming incremental model adaptation. In this work, we propose ELITE, an online CL framework with device-cloud collaboration, to realize on-device real-time inference on time-varying task streams with performance guarantee. To realize on-device real-time inference in online CL, ELITE features a new design of the model zoo comprising various pre-trained models with the assistance of the cloud, and proposes a task-oriented on-device model selection to quickly retrieve the best-fit models instead of performing time-consuming model retraining. To prevent performance degradation on new tasks not available in the cloud, we introduces a latency-aware on-device model fine-tuning strategy to adapt to new tasks with accuracy-latency trade-off, and dynamically updates the model zoo in the cloud to enhance ELITE. Extensive evaluations on five real-world datasets have been conducted, and the results demonstrate that ELITE consistently outperforms the state-of-art solutions, improving the accuracy by 16.3% on average and reducing the response latency by up to 1.98 times.

## 1 Introduction

Nowadays, massive data are continuously collected from ubiquitous end devices, and required immediate process to support real-time data analysis applications, *e.g.*, real-time detection on massive IoT data [54], real-time recommendation in web applications [18] and real-time person identification through surveillance cameras [12]. The time-varying task streams in these applications urges end devices to learn in a continual fashion [15, 58]. Online CL gains increasing interests to learn incrementally from task streams, and much efforts have been proposed to realize stability-plasticity trade-off, meeting that models should learn new tasks (plasticity) while retaining the learned one (stability) [8, 25]. Despite promising, current online CL has primarily focused on optimizing the learning performance, overlooking the requirements of system performance, such as the inference latency and resource efficiency. The performance of real-time inference in online CL deteriorates significantly.

Most of previous efforts in online CL have employed sophisticated and time-consuming model retraining process to guarantee learning performance, making on-device real-time inference infeasible. Specifically, when new tasks arrive, directly utilizing the current model on end devices for inference leads to performance degradation due to changes in data distribution [14, 28]. To avoid this, classical online CL usually conduct the on-device model retraining with new data samples [15, 58]. In this way, it takes a long time to perform model retraining on the new tasks, which would result in delayed model response. Thus, there exist related works proposed to realize online CL on resource-constrained end devices. Most existing practices focus on reducing resource consumption through designing lightweight models or using fewer samples [5, 21, 57, 59, 65]. However, these approaches still struggle to make an effective trade-off between model performance and resource consumption. Therefore, it is highly necessary for resource-constraint end devices to realize real-time inference on time-varying task streams.

There exist two technical challenges to realize on-device real-time inference in online CL. First, end devices are unable to execute the classical model adaptation (*i.e.*, sample replay [31] and model expansion [62]), since there are not substantial computation resources and data samples on end devices [40, 56]. Specifically, compared to the setting with abundant resources, the classical online CL algorithms (*i.e.*, EWC++ [6], GDumb[41] and AGEM[7]) have a noticeable performance drop (about 10%, in Section 2) in the inference accuracy for the new learning task on resource-constrained end devices. Second, the time-consuming model adaptation on resource-constraint end devices makes it not possible to realize real-time inference. When new tasks arrive, classical online CL approaches typically perform model adaptation with new data samples, and then use the upgraded model for inference response. However, in real-world scenarios, new tasks often require immediate inference response without waiting for model adaptation. For example, high-velocity task streams, such as video analysis streams where traffic cameras capture 25 frames per second [23], necessitate real-time inference for vehicle tracking and re-identification to prevent traffic accidents. Compared to short intervals between task arrivals, the model adaptation on resource-constrained end devices is quite computation-intensive, inevitably resulting in prolonged response times. Thus, to realize real-time inference on high-velocity task streams, it is imperative to reduce the computation overhead without performance degradation in online CL on end devices.

To realize real-time inference on resource-constrained end devices, we propose a new d**E**vice-cloud co**L**laborat**I**ve onl**I**ne con**T**inual l**E**arning framework, namely ELITE, which enables end devices to calibrate the on-device model timely with the support of the cloud. ELITE explores the connection between continual (sequential) training and multitask (simultaneous) training, where both of them aim to obtain a solution that performs well across various tasks, and regards multi-task learning (MTL) as the upper bound of CL. In particular, ELITE leverages MTL with abundant cloud-side data resources to pre-train various models for different tasks, forming a model zoo in the cloud. To realize real-time inference on high-velocity and time-varying task streams, ELITE propose a task-oriented online model selection to extract feature with a low compute cost, and retrieve the best-fit models from the model zoo in a fast and robust way. Furthermore, we address the extended scenario where the cloud, lacking new data samples on end devices, is unable to

provide efficient models for real-time inference. To enhance ELITE, we propose the latency-aware model fine-tuning on end devices and dynamic model zoo updating in the cloud to adapt to new tasks with accuracy-latency trade-off. With this device-cloud collaboration in ELITE, we avoid the time-consuming incremental model retraining, because the time cost of model selection and model fine-tuning is significantly lower than that of model retraining on resource-constrained end devices.

The main contributions of our work are summarized as follows:

- In this work, we aim to realize real-time inference in online CL on resource-constrained end devices, and propose ELITE, a new device-cloud collaborative CL framework for time-varying task streams.
- To realize on-device real-time inference in online CL, ELITE features a new design of the model zoo comprising various pre-trained models with the assistance of the cloud, and proposes a task-oriented on-device model selection to quickly retrieve the best-fit models from the cloud.
- To prevent performance degradation on new tasks not available in the cloud, we introduces a latency-aware online model fine-tuning strategy to adapt to new tasks with accuracy-latency trade-off, and dynamically updates the model zoo to enhance ELITE.
- Extensive evaluations on five image and video datasets have been conducted, and the results demonstrate that ELITE improves the accuracy by 16.3% on average, and reduces the response latency by up to 1.98×, compared to the state-of-the-art approaches.

## 2 Background and Related Works

In this section, we provide a comprehensive review on online CL, real-time inference and device-cloud collaboration.

### 2.1 Online Continual Learning

Online CL has garnered increasing attention for its ability to learn incrementally from data streams, by enabling frequent model re-training to adapt to new arriving data samples, and not forgetting the previously learned knowledge [9, 16]. The current online CL approaches to overcome catastrophic forgetting can be identified into three main categories: parameter regularization [1, 27, 32], sample replay [24, 31, 45] and model expansion [19, 34, 62]. These approaches require substantial computation resources and data samples to perform model adaptation, rendering them unsuitable for resource-constrained end devices. As shown in Figure 1(a), we have evaluated five representative CL algorithms deployed on Jetson Nano [52], a end device from INVIDIA, and observe that these algorithms have high learning performance with unrestricted computation resources, while the model performance of on-device CL degrades significantly in resource-limited scenarios. Specifically, the corresponding model performance decreases about 10% with 50% computation resources in use. Moreover, as stated in computationally budgeted CL [40], all existing CL approaches, including distillation [33], sampling [3, 48], FC layers correction [17] and model expansions [42], fail to have good model performance in

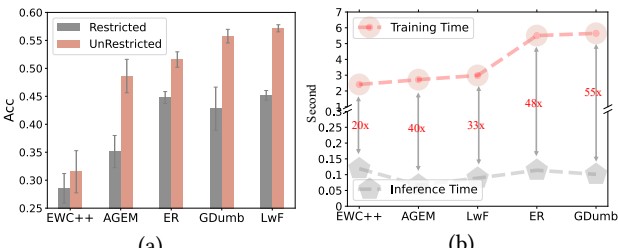

Figure 1: (a) the performance comparison of different resources in use on the Jetson Nano; (b) the time comparison of model training and inference on the Jetson Nano.

Table 1: The communication latency of model transmission by using six classical models with different size.

| Model | Size | Training Time | Comm Latency |
|---|---|---|---|
| CNN | 0.304MB | 1.401s | 0.0026s |
| LeNet5 | 2.181MB | 2.021s | 0.0043s |
| SqueezeNet | 2.869MB | 3.850s | 0.0114s |
| ShuffleNet V2 | 8.772MB | 5.020s | 0.0384s |
| MobileNet V2 | 13.501MB | 5.331s | 0.0819s |
| ResNet18 | 42.838MB | 6.577s | 0.1295s |

a computation-constraint setting. Therefore, it is critical to realize online CL with performance guarantee under the constraint of computation resources.

### 2.2 Real-Time Inference

Massive task streams are required real-time inference to support time-sensitive intelligent applications [12, 14, 38]. However, most of online CL approaches with time-consuming model adaptation makes on-device real-time inference infeasible. As shown in Figure 1(b), considering different CL algorithms (i.e., EWC++ [6], AGEM[7], LwF[35], ER[46] and GDumb[41]), the time consumption of model adaptation is up to 55 times of that of model inference, which would result in a long-time model adaptation before conducting the model inference for the current task. There exist several related works proposed to realize timely model inference on task streams. In order to realize inference queries at any time, Koh Hyunseo *et al.* design a new memory management scheme and learning rate scheduling strategy to adapt to online blurry task streams [28]. To evaluate current CL methods, a new real-time evaluation in online CL has been proposed to take the delay of model training and change in data distribution into account [14]. Although these methods can perform model inference without any delay, the performance of real-time inference is subpar due to the reuse of an out-of-date model, especially for the CL methods with high model retraining cost [2, 41], and task streams with fast data distribution change and high throughout. Thus, it is crucial to achieve real-time inference in online CL while ensuring learning performance guarantees.

### 2.3 Device-Cloud Collaboration

The new paradigm of device-cloud collaborative learning is emerging to leverage the advantages of both end devices and the cloud

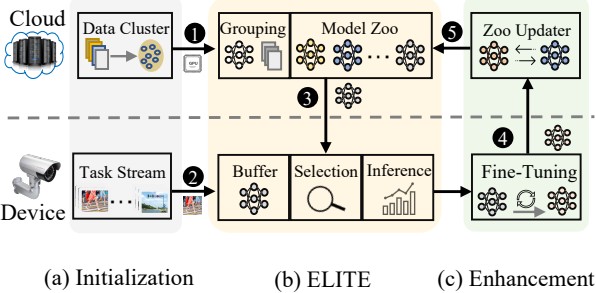

(a) Initialization    (b) ELITE    (c) Enhancement

**Figure 2: The overview of device-cloud collaboration.**

server [55, 60]. Most of previous efforts aims to offload the computation intensive tasks on end devices to the cloud [36, 37, 63]. However, these methods require uploading a significant amount of raw data with considerable communication latency. While AMS [50] and DCCL [61] have tried to unload partial computation to the cloud to alleviate the computation deficiency on end devices, they primarily focus on the optimization of model training and aggregation, which are not applicable to realize real-time inference. Comparing to previous efforts, we prefer to enable real-time inference on resource-constrained end devices by retrieving suitable models from the cloud with model transmission. However, it is noteworthy that device-cloud collaboration may incur communication latency due to model transmission. As shown in Table 1, we measure the communication latency of model transmission by using six classical models (*i.e.*, LeNet5, SqueezeNet, CNN, ShuffleNet V2, MobileNet V2 and ResNet18) with different sizes . We find that it is feasible to realize cloud-enabled on-device CL, as the communication latency is extremely short comparing with the time cost of model adaptation. Despite promising, realizing cloud-enabled on-device CL for real-time inference also poses new challenges. Although we can resolve the problem of insufficient computation resources on end devices by requesting models from the cloud, the separation between arriving task streams on end devices and machine learning models in the cloud makes it difficult to retrieve suitable models for end devices. Moreover, the mismatch between outdated models in the cloud and the high-velocity task streams with frequent data distribution variations on end devices undermines the model performance of real-time inference. Therefore, it is urgent to realize real-time inference in online CL with efficient device-cloud collaboration.

## 3 Overview of Device-Cloud Collaboration

As shown in Figure 2, to realize real-time inference on resource-constraint end devices with device-cloud collaboration may involve the following three stages: (a) Initialization: This stage serves as the preparation for model zoo generation and real-time inference. It involves the clustering of massive data for multi-task model training in the cloud, coupled with the establishment of task streams with frequent data distribution variations on end devices; (b) ELITE: This is our primary design to realize real-time inference with two main components: the cloud-enabled model zoo and on-device

real-time inference. The cloud-enabled multitask model zoo is an offline component that pretrains and stores a collection of multi-task models in the cloud to handle inference requests from end devices. The on-device real-time model inference is an online component responsible for task-oriented model selection, aiming to identify the best-fit models from the model zoo in the cloud instead of time-consuming model retraining; (c) Enhancement: To prevent the performance degradation of ELITE when the cloud is unable to provide efficient models, we propose the latency-aware model fine-tuning on end devices, and dynamic model zoo updating in the cloud to adapt to new tasks with accuracy-latency trade-off.

The process of device-cloud collaboration can be summarized as follows: First, we ❶ segment the entire dataset in the cloud server into various data clusters, associate each data cluster with a corresponding skilled model, and then form the model zoo. It is worth to note that each data cluster can capture the data distribution from multiple tasks, and thus the corresponding model can also handle the inference requests from multiple tasks. When task streams ❷ arrive, end devices need to determine whether to utilize local models in buffer or to request suitable models from the cloud. If the local models on end devices can effectively handle the current tasks, we use these local models to perform real-time inference without additional operations. Otherwise, we ❸ perform an online model selection to request several multi-task models from the model zoo, and the cloud server then transmits the corresponding multi-task models to end devices for real-time inference. After completing real-time inference on current tasks, if the models requested from the cloud prove inefficient for new tasks, end devices ❹ must perform model fine-tuning to adapt to new data samples, and subsequently transmit the fine-tuned model back to the cloud server. With the newly fine-tuned models collected from end devices, the zoo updater ❺ replaces outdated multi-task models to improve the plasticity of model zoo.

## 4 Design of ELITE

In this section, we provide the design of two components in ELITE: the cloud-enabled model zoo and on-device real-time inference.

### 4.1 Cloud-Enabled Model Zoo

To generate a multi-task model zoo, we first use the k-means algorithm to cluster the entire data samples on the cloud server into $n$ data clusters. The data samples in each cluster with high similarity are regarded as a training task. For the set of $n$ training tasks $\mathcal{T} = \{\tau_1, \tau_2, ..., \tau_n\}$, the objective of multi-task training is to identify the model $\theta$ that minimizes the average loss across $n$ tasks:

$$\min_{\theta} \mathcal{L}(\theta, \mathcal{T}) = \frac{1}{n} \sum_{i=1}^{n} \mathcal{L}_i(\theta, \tau_i), \tag{1}$$

where $\mathcal{L}_i$ represents the loss associated with task $\tau_i$, and $\mathcal{L}$ denotes the average loss across $n$ tasks. Considering the task competition and model capability in multi-task learning, it is inefficient to train a single multi-task model with all $n$ tasks [53, 64]. Therefore, we prefer to construct a zoo of $m$ ($m < n$) multi-task models $\Theta = \{\theta_1, \theta_2, ..., \theta_m\}$, such that each model $\theta_i \in \Theta$ can handle a subset of $n$ tasks, thereby ensuring inference performance. In this manner, we first compute the affinity score $Z_{ij}$ to characterize the task

relationship between task $\tau_i$ and $\tau_j$ as follows:

$$Z_{ij} = \frac{\mathcal{L}_i(\theta_{ij}, \tau_i)}{\mathcal{L}_i(\theta_i, \tau_i)} + \frac{\mathcal{L}_j(\theta_{ji}, \tau_j)}{\mathcal{L}_j(\theta_j, \tau_j)}, \forall i \neq j, i, j \in [1, n], \quad (2)$$

where $\frac{\mathcal{L}_i(\theta_{ij}, \tau_i)}{\mathcal{L}_i(\theta_i, \tau_i)}$ represents the affinity of task $\tau_j$ with respect to task $\tau_i$, and similar with $\frac{\mathcal{L}_j(\theta_{ji}, \tau_j)}{\mathcal{L}_j(\theta_j, \tau_j)}$. When the affinity score $Z_{ij}$ is lower, it is more efficient to group these two tasks together for multi-task models training.

Without the information of task streams on end devices, the multi-task models pretrained in advance may be unsuitable for on-device model inference. To enhance the plasticity of multi-task models in model zoo, it is crucial to maximize the diversity of tasks that the pretrained multi-task model involve with. Information entropy has been employed to incorporate diversity and is also widely used as a diversity index [20, 44]. Thus, the problem of task grouping for model zoo generation can be formulated as follows:

$$\max_X H(X) = -\sum_{j=1}^{m} \sum_{i=1}^{n} \frac{X_{ij}}{\sum_{k=1}^{n} X_{kj}} \log \frac{X_{ij}}{\sum_{k=1}^{n} X_{kj}} \quad (3)$$

$$s.t. \sum_{i=1}^{n} \sum_{k=i+1,}^{n} X_{ij} \cdot X_{kj} \cdot Z_{ik} \leq a, \quad \forall j \in [1, m], \quad (3a)$$

$$\sum_{j=1}^{m} X_{ij} \leq b_i, \quad \forall i \in [1, n], \quad (3b)$$

$$X_{ij} \in \{0, 1\}, \quad \forall i \in [1, n], j \in [1, m], \quad (3c)$$

where $X_{ij}$ is the decision variable indicating whether the task $\tau_i$ is assigned to the model $\theta_j$, and $H(X)$ denotes the information entropy of grouped task sets. The first constraint (3a) ensures that the multiple tasks utilized for model training are mutually beneficial by requiring the sum of their affinity score less than a given threshold $a$. The second constraint (3b) ensures that each task $\tau_i \in \mathcal{T}$ can be allocated at most $b_i \in \mathcal{B}$ times where $\mathcal{B} = \{b_1, b_2, ..., b_n\}$. The last constraint (3c) indicates that $X_{ij}$ is a binary decision variable. We can find that this optimization problem of task grouping can be reduced to a multi-dimensional binary knapsack problem [4, 11], which is known to be NP-hard. Specifically, the objective of this optimization problem is to maximize the diversity of tasks by selecting the binary option under two capacity constraints.

Traditional algorithms for solving the multi-dimensional knapsack problem, such as dynamic programming or branch-and-bound methods, are often computationally expensive and struggle to scale efficiently with increasing problem dimensions and complexity. Instead of exhaustively searching the solution space, ELITE introduces a heuristic greedy allocation strategy with sequential two-step allocation: an initial random allocation to satisfy the second constraint in Eq. (3b), following a greedy reallocation to adjust the initial solution to meet the first constraint in Eq. (3a). As shown in Algorithm 1, ELITE first generates an initial solution to satisfy the second constraint in Eq. (3b) in a random manner, by regarding the training task $\tau_i$ allocated at most $b_i$ times as the same $b_i$ training tasks (line 1 and 2). According to the first constraint in Eq. (3a), ELITE divides the the initial random allocations into two sets: the satisfied task allocation set $\mathcal{S}$ and the unsatisfied set $\mathcal{U}$ (line 3). To adjust the initial solution (line 4 to 13), ELITE employs a greedy

---

**Algorithm 1:** Cloud-Enabled Model Zoo

**Input:** $n$ training tasks $\mathcal{T} = \{\tau_1, ..., \tau_n\}$, affinity scores $\{Z_{ij}\}$, budget $\mathcal{B} = \{b_1, ..., b_n\}$, threshold $a$

**Output:** model zoo $\Theta = \{\theta_1, \theta_2, ..., \theta_m\}$

1   For each $j \in [1, m], \mathcal{T}_j \leftarrow \emptyset$;

2   $\{\mathcal{T}_j\}_{j=1}^{m} \leftarrow$ Randomly allocate $X$ with $\mathcal{B}$;

3   $\mathcal{S}, \mathcal{U} \leftarrow$ Divide $\{\mathcal{T}_j\}_{j=1}^{m}$ by using Eq. (3a);

4   **while** $|\mathcal{U}| > 0$ **do**

5     **for** $\mathcal{T}_j \in \mathcal{U}$ **do**

6       $i \leftarrow arg\max_{i \in \mathcal{T}_j} \{\sum_{k \neq i, k \in \mathcal{T}_j} Z_{i,k}\}_{i=1}^{|\mathcal{T}_j|}$;

7       **for** $\mathcal{T}_k \in \mathcal{S}$ **do**

8         $a_k \leftarrow$ Compute $\mathcal{T}_k \cup \{i\}$ by using Eq. (3a);

9         Compute $\triangle H_k$ by using Eq. (4), if $a_k < a$;

10      $k \leftarrow arg\max_{k, \mathcal{T}_k \in \mathcal{S}} \{\triangle H_k\}$;

11      $\mathcal{T}_j \leftarrow \mathcal{T}_j \setminus \{i\}, \mathcal{T}_k \leftarrow \mathcal{T}_k \cup \{i\}$;

12      $a_i \leftarrow$ Compute $\mathcal{T}_j$ by using Eq. (3a);

13      $\mathcal{U} \leftarrow \mathcal{U} \setminus \{\mathcal{T}_j\}, \mathcal{S} \leftarrow \mathcal{S} \cup \{\mathcal{T}_j\}$, if $a_j < a$;

14   Pretrain multi-task models $\{\theta_j\}_{j=1}^{m}$ with $\mathcal{S}$;

15   **return** $\Theta = \{\theta_1, \theta_2, ..., \theta_m\}$.

---

reallocation strategy to satisfy the first constraint in Eq. (3a) while maximizing the entropy increment computed as follows:

$$\triangle H_j = H(X_{ij} = 1) - H(X_{ij} = 0), \forall i \in [1, n], j \in [1, m]. \quad (4)$$

ELITE pre-trains multi-task models $\{\theta_j\}_{j=1}^{m}$ in model zoo based on the satisfied task allocation $\mathcal{S}$ (line 14). During this task allocation process, there exist at most $|\mathcal{U}| \leq m$ unsatisfied sets that need to be reallocated, and each set contains at most $n \cdot \max\{b_i\}_{i=1}^{n}$ training tasks to be reallocated. Thus, the time complexity of this algorithm is $O(m \cdot n \cdot \max\{b_i\}_{i=1}^{n})$. This sequential two-step allocation approach obtains an efficient solution with reduced computational cost, and provides better flexibility in addressing complex, multi-constraint problems that existing algorithms might fail to solve efficiently.

## 4.2 On-Device Real-Time Inference

Given the non-stationary task streams $\mathcal{B}_t \sim \mathcal{D}_t$ on end devices, where $\mathcal{D}_t$ is the data distribution at time step $t$, the objective of real-time inference is to obtain a model $\theta^t$ that predicts a label $y^t \in \mathcal{Y}$ for an input feature $x^t \in \mathcal{X}$ without any delay. At each time step $t$, we execute the following two steps to realize real-time model inference: (1) First, the current task reveals the input of current data samples $\{x_i^t\}_{i=1}^{n_t} \subseteq \mathcal{B}_t$, and end devices utilize these data samples to perform model selection from either local models in buffer or the model zoo in the cloud; (2) Then, we employ the selected models to generate predictions $\{\overline{y_i^t}\}_{i=1}^{n_t}$ for the given $\{x_i^t\}_{i=1}^{n_t}$. Upon completing real-time inference, the system reveals the true labels $\{y_i^t\}_{i=1}^{n_t}$ (i.e., generated by large models or human annotators [43]), and evaluate the performance of real-time inference by comparing $\{\overline{y_i^t}\}_{i=1}^{n_t}$ to $\{y_i^t\}_{i=1}^{n_t}$. This process highlights the importance of efficient model selection in achieving real-time inference with guaranteed performance.

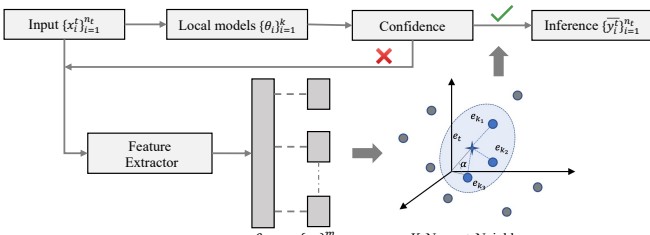

**Figure 3: The illustration of on-device model selection.**

As illustrated in Figure 3, there exist two probabilities for retrieving the best-fit models from the local buffer on end devices or the model zoo in the cloud. To ascertain whether the local models on end devices are adequate for the current tasks, we calculate prediction uncertainty to evaluate the confidence of the local models in predicting the input data [13, 39]. The prediction uncertainty, denoted as $U$, is computed as follows:

$$U = -\sum_{k=1}^{c} \sigma(y = k|x) \log \sigma(y = k|x), \sigma(y = k|x) = \frac{e^{z_k}}{\sum_{i=1}^{c} e^{z_i}} \quad (5)$$

where $\sigma(y = k|x)$ represents the probability of predicting class $k$ with the given input $x$, and $z_k$ denotes its logits of the final layer in the model. If the value of $U$ is small, end devices prefer to utilize the local models for task inference. Otherwise, models are requested from the model zoo in the cloud.

Unlike directly applying neural networks to determine model selection from the cloud without performance guarantee [47], ELITE enables task-oriented on-device model selection by transmitting a pretrained model as feature extractor (*e.g.*, adopting ResNet18 pretrained on ImageNet as feature extractor), and feature embeddings as task representations from the cloud to end devices in advance. We extract features of data samples by using the pretrained model as domain similarity, and aggregate the features of all data samples pretrained in multi-task models to form their task embeddings:

$$E = \{e_i | e_i = \sum_{j=1}^{n_i} \frac{g(x_j)}{n_i}\}_{i=1}^{m}, \quad (6)$$

where $E$ is the set of task embedings of multi-tasks models in model zoo, and $g(x_j)$ represents the feature embedding of the input $x_j$ extracted with the feature layers $g$ of the pretrained model. With the set of task embeddings $E = \{e_i\}_{i=1}^{m}$ provided from the cloud, we extract the current task embedding $e_t$ in the same way, and select the most $k$ suitable multi-task models with the KNN method into the candidate set $R_t$:

$$\min_{R_t \subseteq E} \sum_{e_t \in R_t} \|e_t - e_i\|, \ s.t. \ |R_t| = k. \quad (7)$$

After obtaining the $k$ most suitable models $\{\theta_i^t\}_{i=1}^{k}$ from the cloud server, ELITE selects the model with highest confidence to realize model inference, and evaluates the inference performance on the current task $\mathcal{B}_t$ by revealing the true labels $\{y_i^t\}_{i=1}^{n_t}$:

$$Acc_t = Acc(\theta_i^t, \{y_i^t\}_{i=1}^{n_t}), i = arg \min\{U_i\}_{i=1}^{k}, \quad (8)$$

where $Acc(\theta_i^t, \{y_i^t\}_{i=1}^{n_t})$ is the prediction accuracy of the selected model $\theta_i^t$ on the current task, and $U_i$ denotes its prediction uncertainty. If the inference performance $Acc_t$ is low, it indicates that the current task is new to both end devices and the cloud server. In this scenario, ELITE needs to perform model fine-tuning to adapt to the new task, which will discuss in next.

## 5 Enhancement of ELITE

In this section, we focus on the extension scenario where the cloud cannot provide efficient inference models to enable on-device real-time inference, and introduce latency-aware model fine-tuning and dynamic zoo updating to enhance ELITE.

### 5.1 Latency-aware Model Fine-Tuning

If the accuracy of real-time inference with on-device model selection is low, model fine-tuning is necessary to enhance the performance of inference models without long-time delay. To adapt to new tasks and reduce the time consumption of model fine-tuning, we consider a sparse fine-tuning approach that avoids massive computation, and assume that each inference model has $L$ layers. To determine which layers of the inference model $\theta^t$ should be fine-tuned, we further generate a sparse mask $z_t \in \{0, 1\}^L$, where $z_t^i = 1$ indicates that the $i$-th layer of the inference model $\theta^t$ will be updated; otherwise, it will remain frozen without any operation. Moreover, the value of sparse mask $z_t$ is closely related to the latency constraints $\triangle t$:

$$z_t^i = \begin{cases} 1, & if \ \triangle t - (L - i)\xi > 0 , \\ 0, & otherwise . \end{cases} \quad (9)$$

where $\xi$ represents the time cost of fine-tuning one layer in the inference model.

To improve the performance of inference models on the current task, we need to allocate the given latency $\triangle t$ among the $k$ candidate models $\{\theta_i\}_{i=1}^{k}$. Therefore, we formulate the following time allocation optimization problem:

$$Acc_t = max\{Acc(\theta_i^t(z_t(\triangle t_i)), \{y_i^t\}_{i=1}^{n_t})\}_{i=1}^{k}, s.t. \sum_{i=1}^{k} \triangle t_i = \triangle t, \quad (10)$$

where $\theta_i^t(z_t(\triangle t_i))$ denotes the model $\theta_i^t$ fine-tuned with the allocated time $\triangle t_i$. However, this time allocation problem can be framed as the multi-armed bandits (MAB) due to the uncertainty of $Acc(\theta_i^t(z_t(\triangle t_i)), \{y_i^t\}_{i=1}^{n_t})$. To address this intractable problem, we propose a two stage time allocation method with exploration and exploitation to improve model performance:

$$\begin{aligned} Acc_t &= Acc(\theta_i^t(z_t(\frac{\triangle t}{2})), \{y_i^t\}_{i=1}^{n_t}), \\ s.t. \ i &= arg \max\{Acc(\theta_i^t(z_t(\frac{\triangle t}{2k})), \{y_i^t\}_{i=1}^{n_t})\}_{i=1}^{k}. \end{aligned} \quad (11)$$

Specifically, we divide the given latency into two equal time slots. In the first slot, we allocate $\frac{\triangle t}{2}$ evenly among $k$ candidate models, and perform model fine-tuning with the same amount of time $\frac{\triangle t}{2k}$ to evaluate its model performance. In the second slot, we select the candidate model with highest prediction accuracy in the first slot, and use the remaining time $\frac{\triangle t}{2}$ to continually fine tune this selected model. In this way, we obtain an efficient fine-tuned model to adapt to new data samples.

**Table 2: The details of Datasets. We consider two kind of task streams: image streams and video streams.**

| Dataset | Classes | Samples | Size | Task Stream |
|---------|---------|---------|------|-------------|
| CIFAR10 | 10 | 50k | 170MB | Image Classification |
| CIAFR100 | 100 | 50k | 197MB | Image Classification |
| Tiny-ImageNet | 200 | 100k | 1.1GB | Image Classification |
| HDMB51 | 51 | 6.84k | 2.12GB | Video Analytic |
| UCF101 | 101 | 13.32k | 6.93GB | Video Analytic |

## 5.2 Dynamic Zoo Updating

When the current task is new and not available in the cloud, there is a urgent to dynamically update the multi-task model zoo to improve its plasticity. Traditional approaches tend to upload new data samples to the cloud for model retraining, resulting in significant communication overhead and data privacy concerns. To avoid uploading new data samples, we propose an incremental zoo updater designed to refresh multi-task models in a timely and efficient manner. At each time step, the incremental zoo updater dynamically enhances model zoo with fine-tuned models collected from end devices. To determine which model to replace, the zoo updater monitors the request counts for each multi-task model in the model zoo, identifying the least requested model for replacement. In this way, the model zoo updater can effectively learn the changing patterns of task streams on end devices, retains useful models in the model zoo, thereby enhancing the efficiency of on-device model selection and avoiding unnecessary model fine-tuning on end devices.

## 6 Experiment

In this section, we first describe the experimental setup, and then report the experimental results with the performance analysis.

### 6.1 Experiment Setup

**Datasets and Tasks**. We consider two kind of task streams including image classification and video analytic with five datasets as shown in Table 2. For image classification tasks, we utilize three different image datasets: CIFAR10/100 [29] and Tiny-ImageNet [10]. In video analytic tasks, we consider two classical video datasets: HDMB51 [30] and UCF101 [51]. As for task streams, we consider class-incremental continual learning [26, 49] by dividing the whole classes of each dataset into different task groups. Moreover, we consider two types of task streams: fuzzy-boundary and sharp-boundary task stream. In fuzzy-boundary task stream, the classes of each task are randomly selected, and adjacent tasks may have the same classes. As for sharp-boundary task stream, adjacent tasks are different, indicating no overlapping classes.

**Baselines**. The current practices about online CL can be categorized into two main categories: cloud-enabled online CL approaches and on-device one. The cloud-enabled online CL performs model adaptation on the cloud server and model inference on end devices, *i.e.*, AMS [50] and RECL [22]. As for on-device CL, both model training and inference are conducted entirely on end devices, *i.e.*, EWC++ [6], LwF[35], GDumb[41], A-GEM[7], ER[46] and MIR[2].

**Evaluation Metrics**. We evaluate the performance of real-time inference in Online CL by using the following three metrics:

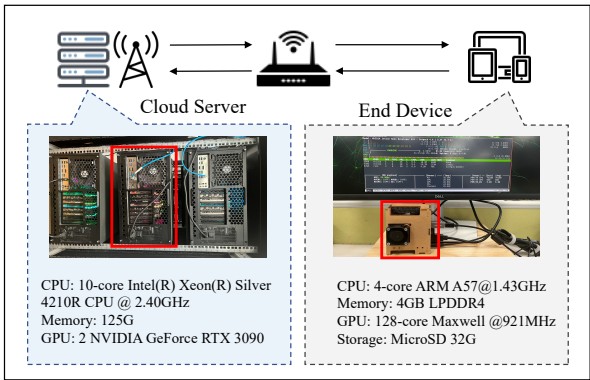

**Figure 4: The prototype system of ELITE.**

- Average Accuracy: Given the length of task streams $T$, we use $\triangle f(x_t, y_t; \theta)$ to denote the prediction accuracy of predictor $\theta$ where $x_t$ is the input feature, $y_t$ is the truth label and $t$ is the time step of the current task. The average accuracy at the end of task streams can be measured as:

$$\mathcal{A} = \frac{1}{T} \sum_{t=1}^{T} \triangle f((x_t, y_t; \theta)). \tag{12}$$

- Response Latency: Given the length of task streams $T$, we compute the inference time $c_t$ and the waiting time $\tau_t$ (*e.g.*, the time of model retraining or fine-tuning) at time step $t$, and add these two time cost as its response latency $\mathcal{L}_t$. The response latency of task streams can be obtained as:

$$\mathcal{L} = \frac{1}{T} \sum_{t=1}^{T} \mathcal{L}_t = c_t + \tau_t. \tag{13}$$

- Forgetting Rate: It evaluates how much the model forgets about the previously learned knowledge, and measured by comparing the prediction accuracy at the end of task streams with the one at the beginning in task streams:

$$\mathcal{F} = \frac{\max_i(\triangle f(x|t=1, y) - \triangle f(x|t=i, y))}{\triangle f(x|t=1, y)}. \tag{14}$$

**Implementation Details**. To conduct extensive experiments, we select five lightweight models: LeNet5, SqueezeNet, ShuffleNet V2, MobileNet V2 and ResNet18. The learning rate and batch size of model retraining are 0.01 and 64, respectively. Moreover, we set the latency for model fine-tuning to one second, *i.e.*, $\triangle t = 1s$. To make the results more persuasive, all experiments are performed three times. As shown in Figure 4, we realize the prototype system of ELITE by utilizing a Jetson Nano with 4GB memory and the cloud server with 2 NVIDIA RTX 3090. The communication interaction between Jetson Nano and the cloud server is facilitated through wifi routers. All CL methods are written in Python with PyTorch.

### 6.2 Overall Performance

Table 3 analyzes average accuracy, response latency, and forgetting for CL methods across five datasets: CIFAR10/100, Tiny-ImageNet, HDMB51, and UCF101. ELITE consistently shows the highest inference performance in image classification tasks (CIFAR10/100,

Table 3: The performance comparison of different CL methods on five different datasets.

| | | EWC++ | MIR | LwF | ER | AGEM | GDumb | AMS | RECL | ELITE |
|---|---|---|---|---|---|---|---|---|---|---|
| CIFAR10 | $\mathcal{A}$ | 0.176 ± 0.063 | 0.268 ± 0.093 | 0.275 ± 0.028 | 0.371 ± 0.030 | 0.130 ± 0.016 | 0.277 ± 0.004 | 0.125 ± 0.021 | 0.172 ± 0.002 | 0.413 ± 0.039 |
| | $\mathcal{L}(s)$ | 2.011 ± 0.488 | 3.031 ± 0.518 | 1.496 ± 0.292 | 2.589 ± 0.220 | 2.576 ± 0.138 | 3.268 ± 0.291 | 1.971 ± 0.034 | 1.441 ± 0.097 | 1.127 ± 0.201 |
| | $\mathcal{F}$ | 0.844 ± 0.063 | 0.736 ± 0.093 | 0.581 ± 0.030 | 0.851 ± 0.016 | 0.742 ± 0.028 | 0.778± 0.004 | 0.881 ± 0.021 | 0.791 ± 0.001 | 0.581 ±0.039 |
| CIFAR100 | $\mathcal{A}$ | 0.176 ± 0.023 | 0.174 ± 0.029 | 0.153 ± 0.008 | 0.197 ± 0.029 | 0.178 ± 0.023 | 0.051 ± 0.006 | 0.059 ± 0.011 | 0.164 ± 0.011 | 0.397 ± 0.014 |
| | $\mathcal{L}(s)$ | 3.334 ± 0.566 | 6.884 ± 0.108 | 4.078 ± 0.097 | 5.367 ± 0.492 | 7.033 ± 0.138 | 7.207 ± 0.034 | 4.241 ± 0.343 | 1.884 ± 0.199 | 1.341 ± 0.117 |
| | $\mathcal{F}$ | 0.796 ±0.028 | 0.821 ±0.027 | 0.848 ±0.016 | 0.795 ±0.032 | 0.954± 0.039 | 0.792 ±0.026 | 0.921 ±0.015 | 0.834 ±0.013 | 0.587 ±0.056 |
| Tiny-ImageNet | $\mathcal{A}$ | 0.182 ± 0.053 | 0.176 ± 0.034 | 0.207 ± 0.027 | 0.175 ± 0.049 | 0.188 ± 0.060 | 0.101 ± 0.047 | 0.117 ± 0.051 | 0.196 ± 0.038 | 0.275 ± 0.028 |
| | $\mathcal{L}(s)$ | 1.718 ± 0.225 | 3.538 ± 0.632 | 1.589 ± 0.213 | 3.764 ± 0.838 | 2.861 ± 0.562 | 6.074 ± 0.487 | 3.064 ± 0.567 | 1.945 ± 0.474 | 1.034 ± 0.059 |
| | $\mathcal{F}$ | 0.890± 0.042 | 0.881± 0.027 | 0.811 ±0.012 | 0.861± 0.034 | 0.846± 0.044 | 0.895± 0.012 | 0.958 ±0.032 | 0.827± 0.029 | 0.728 ±0.018 |
| HDMB51 | $\mathcal{A}$ | 0.157 ± 0.152 | 0.220 ± 0.136 | 0.346 ± 0.129 | 0.362 ± 0.129 | 0.148 ± 0.146 | 0.192 ± 0.153 | 0.136 ± 0.143 | 0.543 ± 0.130 | 0.654 ± 0.043 |
| | $\mathcal{L}(s)$ | 3.006 ± 0.777 | 4.117 ± 0.761 | 2.482 ± 0.476 | 3.278 ± 0.723 | 3.694 ± 0.419 | 2.884 ± 0.327 | 6.269 ± 2.253 | 1.123 ± 0.263 | 1.032 ± 0.067 |
| | $\mathcal{F}$ | 0.952± 0.016 | 0.771± 0.071 | 0.675 ±0.019 | 0.556 ±0.016 | 0.952± 0.017 | 0.965 ±0.044 | 0.954 ±0.008 | 0.563± 0.047 | 0.328± 0.013 |
| UCF101 | $\mathcal{A}$ | 0.129 ± 0.153 | 0.392 ± 0.138 | 0.252 ± 0.135 | 0.483 ± 0.126 | 0.131 ± 0.149 | 0.188 ± 0.158 | 0.136 ± 0.143 | 0.412 ± 0.106 | 0.652 ± 0.075 |
| | $\mathcal{L}(s)$ | 2.846 ± 0.431 | 4.509 ± 1.022 | 2.519 ± 0.232 | 3.412 ± 0.728 | 3.593 ± 0.955 | 2.994 ± 0.593 | 6.269 ± 2.253 | 1.139 ± 0.258 | 1.033 ± 0.078 |
| | $\mathcal{F}$ | 0.923 ±0.034 | 0.483± 0.081 | 0.831 ±0.020 | 0.424± 0.050 | 0.921 ±0.034 | 0.989 ±0.050 | 0.954 ±0.008 | 0.565± 0.038 | 0.376± 0.066 |

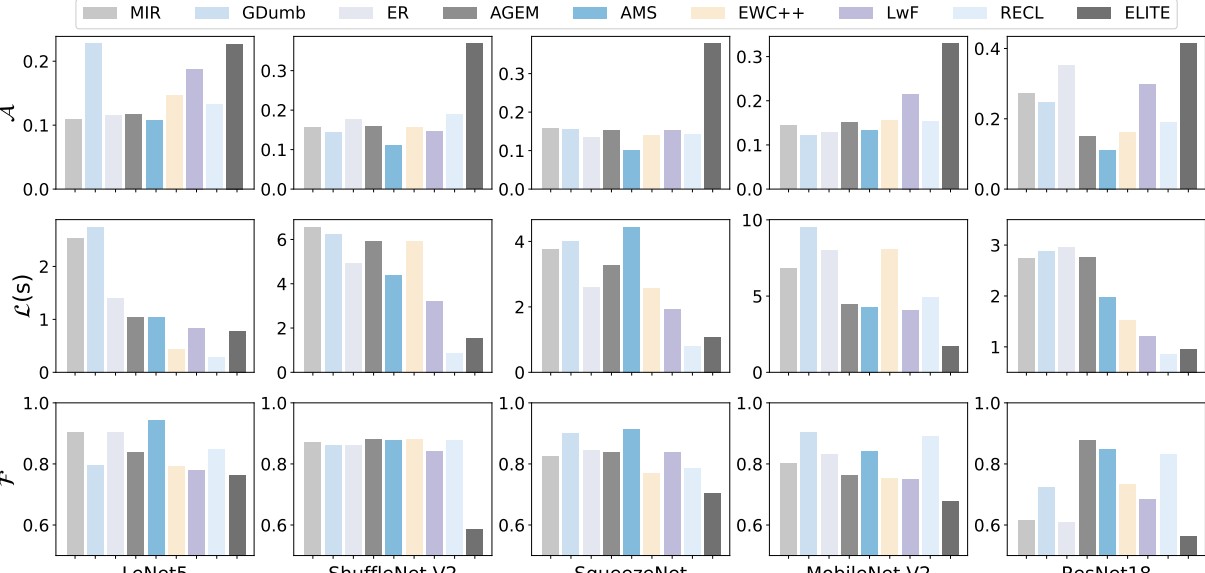

Figure 5: The performance comparison of different CL methods with five different models.

Tiny-ImageNet), though accuracy and forgetting rate decline with increasing data complexity, especially for Tiny-ImageNet. Response latency is affected by both dataset complexity and model adaptation strategy. In video analytics tasks, ELITE maintains around 60% accuracy, benefiting from fewer classes and frame similarity. ELITE outperforms other methods in accuracy, latency, and forgetting across all datasets, demonstrating its reliability.

To validate the stability of the experimental results, we conducted ablation studies on five different models: LeNet5, ShuffleNet V2, SqueezeNet, MobileNet V2, and ResNet18. Figure 5 shows the analysis of average accuracy, response latency, and forgetting rate. While LeNet5 demonstrates the highest accuracy for ELITE among CL methods, its overall performance is low due to its limited capacity. ShuffleNet V2, SqueezeNet, and MobileNet V2 show similar inference performance, with approximately 7% improvement over LeNet5. ResNet18 exhibits the highest inference performance due

to its larger size. ELITE consistently outperforms other CL methods in average performance and response latency. Notably, average accuracy improves with increasing model complexity, while latency remains stable, as ELITE does not require model retraining. In contrast, other CL methods show significant fluctuations in accuracy and latency, further validating the reliability and stability of ELITE.

### 6.3 Robustness of ELITE

Figure 6(a) illustrates the impact of stream type on average accuracy by analyzing both fuzzy-boundary and sharp-boundary task streams. In fuzzy-boundary task streams, ELITE demonstrates an approximately 11% improvement in accuracy compared to other CL methods. In sharp-boundary task streams, ELITE achieves the highest average accuracy among all CL methods, with a significant 16% improvement, indicating its robust performance stability across different task scenarios. Additionally, we observe that most CL

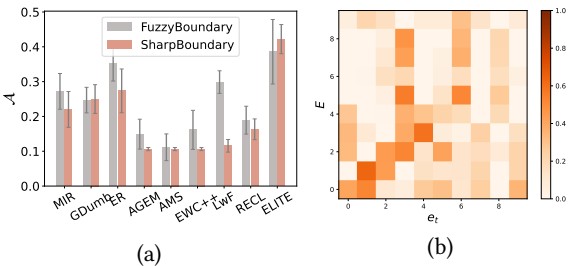

(a)

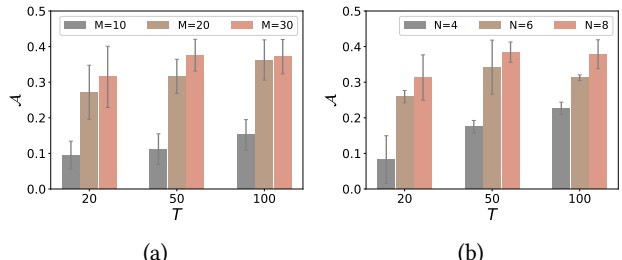

(a)                                    (b)

(b)

Figure 6: The performance comparison on: (a) different task streams; (b) the similarity of feature embeddings.

Figure 8: The performance of model zoo on: (a) different model number; (b) different task number.

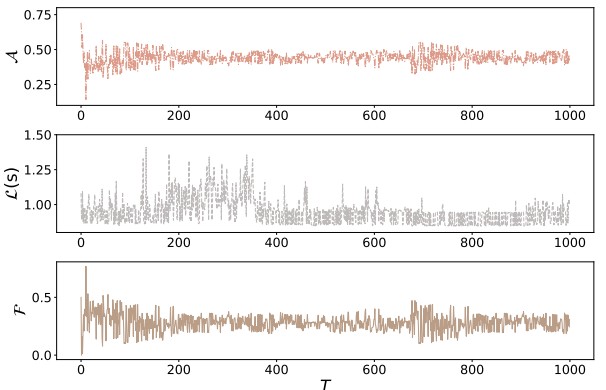

Figure 7: The performance of ELITE with 1000 tasks.

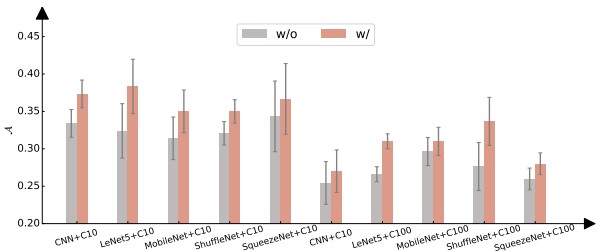

Figure 9: The performance comparison of ELITE with (w/) and without (w/o) the enhancement by using five different models on two datasets.

methods tend to have higher average accuracy in fuzzy-boundary streams compared to sharp-boundary streams, while ELITE exhibits superior performance in sharp-boundary task streams compared to fuzzy-boundary ones, likely because multi-task models often perform better across a broader range of tasks. The performance of feature embedding similarity calculations, as depicted in Figure 6(b), demonstrate that the task-oriented model selection is robust with feature embedding similarity. Figure 7 presents the average accuracy, latency and forgetting rate of ELITE as the length of task streams extends to 1,000 tasks. ELITE maintains stable performance as the number of tasks increases. Initially, average accuracy and forgetting rate show significant fluctuations, but it stabilizes as the number of tasks grows. This behavior can be attributed to the need for continual updates in the model zoo during the early stages to accommodate new tasks.

In Figure 8(a), we denote the number of multi-task models in the zoo as $M$. The results show that increasing the number of multi-task models in the zoo improves average inference accuracy, with a marked improvement at $M = 20$. However, when $M = 30$, accuracy plateaus and performance becomes unstable, indicating that the optimal number of models should be carefully balanced. Next, we analyze the impact of the task number of multi-task models involve with, represented as $N$, on inference performance. The Figure 8(b) shows that as $N$ increases, so does the average accuracy. However, once $N$ reaches a higher value, the rate of accuracy improvement diminishes significantly. This observation underscores the need to

carefully design the task number to optimize model performance. As shown in Figure 9, we validate the performance of ELITE with (w/) and without (w/o) the enhancement by using five lightweight models (*e.g.*, CNN, LeNet, SqueezeNet, ShuffleNet and MobileNet) and two datasets, *e.g.*, CIFAR10 (C10) and CIFAR100 (C100). It is obvious that the inference performance of ELITE with the enhancement is superior to that without additional operations, due to the model fine-tuning to adapt to new tasks.

## 7 Conclusion

In this paper, we focused on the real-time inference on resource-constraint end devices in online CL, and proposed a new device-cloud collaborative CL framework, namely ELITE, for time-varying task streams. To realize real-time model inference, ELITE formed model zoo in the cloud server, and proposed task-oriented on-device model selection on end devices. To prevent performance degradation on new tasks not available in the cloud, we introduced latency-aware online model fine-tuning strategy to adapt to new tasks, and dynamically updated model zoo to enhance ELITE. Extensive evaluations on five image and video datasets have been conducted, and the results demonstrate that ELITE improves 16.3% inference performance and reduces up to 1.98x response latency compared to the-state-of-art solutions.

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
