# OpenReview forum: "Enabling Real-Time Inference in Online Continual Learning via Device-Cloud Collaboration"
_ACM.org/TheWebConf/2025/Conference — WWW 2025 Poster_

### Official Review · Reviewer_DriV · 2024-11-28

**Novelty:** 6
**Technical Quality:** 6

**Review:**

Quality

Pros:
- The paper presents a well-structured and comprehensive approach to addressing the challenges of real-time inference in online continual learning (CL) on resource-constrained devices. ​
- Extensive evaluations on five real-world datasets demonstrate the effectiveness of the proposed ELITE framework, showing significant improvements in accuracy and response latency compared to state-of-the-art solutions. ​
- The methodology is detailed, with clear explanations of the model zoo, task-oriented model selection, and latency-aware model fine-tuning strategies. ​

Cons:
- The paper could benefit from more detailed explanations of certain technical aspects, such as the specific algorithms used for model selection and fine-tuning.
- The complexity of the proposed framework might make it challenging for practitioners to implement without additional guidance or code examples.

Clarity

Pros:

- The paper is well-organized, with a logical flow from the introduction to the experimental results and conclusions.
- Key concepts and methodologies are explained clearly, making it accessible to readers with a background in machine learning and continual learning.

Cons:

- The use of technical jargon without sufficient explanation might be challenging for readers who are not experts in the field.

Originality

Pros:

- The concept of device-cloud collaboration for real-time inference in online CL is innovative and addresses a significant gap in the current research. ​
- The introduction of a model zoo and task-oriented model selection is a novel approach to mitigating the challenges of real-time inference on resource-constrained devices. ​

Cons:

- While the approach is original, it builds on existing concepts in multi-task learning and model fine-tuning, which might limit the perceived novelty for some readers.

Significance

Pros:

- The proposed framework has the potential to significantly impact the field of online CL by enabling real-time inference on resource-constrained devices, which is crucial for many real-world applications. ​
- The improvements in accuracy and response latency demonstrated in the experiments highlight the practical benefits of the ELITE framework. ​

Cons:

- The practical implementation of the framework in real-world scenarios might face challenges related to communication latency and the availability of cloud resources. ​

Summary

Pros:

- Comprehensive and well-structured approach.
- Significant improvements in accuracy and response latency. ​
- Clear explanations and logical flow.
- Innovative concept of device-cloud collaboration. ​
- Potential for significant impact in real-world applications. ​

Cons:

- Some technical aspects could be explained in more detail.
- Complexity might hinder practical implementation. ​
- Use of technical jargon without sufficient explanation.
- Builds on existing concepts, which might limit perceived novelty.
- Practical implementation challenges related to communication latency and cloud resources. ​

Overall, the paper presents a high-quality, original, and significant contribution to the field of online continual learning, with clear explanations and a well-structured methodology. However, it could benefit from more detailed technical explanations and considerations of practical implementation challenges.

**Questions:**

Implementation Details:

- Could you provide more detailed implementation guidelines or code examples for the ELITE framework? This would help practitioners better understand and implement your approach.


Communication Latency:

- How does the communication latency between the device and the cloud impact the overall performance of the ELITE framework in real-world scenarios? ​ Are there any strategies to mitigate potential delays?



Scalability:

- How does the ELITE framework scale with an increasing number of tasks and models in the model zoo? Are there any performance bottlenecks or limitations that need to be addressed? ​


Resource Constraints:

- How does the framework handle extreme resource-constrained environments where both computational power and network bandwidth are limited? ​ Are there any specific optimizations or fallback mechanisms in place?


Real-World Applications:

- Have you tested the ELITE framework in any real-world applications or deployment scenarios? If so, can you share the results and any insights gained from these experiments?


Task Stream Variability:

- How does the framework handle highly variable task streams with frequent and unpredictable changes in data distribution? ​ Are there any mechanisms to quickly adapt to such changes? ​


User Feedback and Adaptation:

- Is there any mechanism for incorporating user feedback or human-in-the-loop interactions to improve model performance and adaptability in real-time?


Security and Privacy: ​

- How does the framework address security and privacy concerns, especially when transmitting data and models between the device and the cloud? ​


Future Work:

- What are the next steps or future directions for this research? Are there any planned improvements or extensions to the ELITE framework? ​

**Reviewer Confidence:**

1: The reviewer's evaluation is an educated guess

**Scope:**

4: The work is relevant to the Web and to the track, and is of broad interest to the community

---

### Official Review · Reviewer_4DGH · 2024-11-29

**Novelty:** 4
**Technical Quality:** 4

**Review:**

This study focused on the real-time inference on resource-constrained end devices in online CL and suggested a novel device-cloud collaborative CL framework, and the system name is ELITE, for time-varying task streams. To implement real-time model inference, ELITE established a model zoo on the cloud server and offered task-oriented on-device model selection for end devices. To avoid performance degradation on new jobs inaccessible in the cloud, the authors proposed a latency-aware online model fine-tuning technique to adapt to new workloads and a dynamically updated model zoo to improve ELITE. Some assessments on five image and video datasets have been undertaken, and the authors' findings claimed that ELITE increases 16.3% inference performance and decreases up to 1.98x response latency compared to SOTA solutions.

**Questions:**

1. The authors present a significant amount of data. Nonetheless, further explanation should be provided to give signifiers to the audiences. Nonetheless, I spotted some interesting results in the paper.
2. In Figure 8, according to the author's claim, when 𝑀 = 30, the accuracy plateaus and the performance becomes unstable, indicating that the optimal number of models should be carefully balanced. Does this result indicate any limitations of the model? Can the author do further elaboration?
3. Some formatting issues, e.g., 1.98X in the introduction but 1.98x in the conclusion - consistency matters.

**Reviewer Confidence:**

3: The reviewer is confident but not certain that the evaluation is correct

**Scope:**

3: The work is somewhat relevant to the Web and to the track, and is of narrow interest to a sub-community

---

### Official Review · Reviewer_ZSRt · 2024-12-03

**Novelty:** 6
**Technical Quality:** 6

**Review:**

This work addresses the challenge of real-time inference in online continual learning (CL), where frequent data variations and model adaptation lead to performance degradation. The proposed framework, ELITE, enables on-device real-time inference through device-cloud collaboration, utilizing a model zoo of pre-trained models and task-oriented model selection. It introduces latency-aware fine-tuning for new tasks and dynamically updates the cloud model zoo. Evaluations on real-world datasets show ELITE improves accuracy by 16.3% and reduces latency by up to 1.98 times over state-of-the-art solutions.

Strengths:
1. The studied problem of real-time Inference of online continual learning on resourced-limited edge devices sounds new and interesting.
2. The presented idea, in terms of using cloud models to calibrate the device models which suffer from drift issues also sound new and interesting.
3. Realistic prototype implementation.
4. Various datasets and benchmarks are adopted to extensively evaluate the performance of the presented solution.

Weakness: see the detailed questions.

**Questions:**

1. The context of the studied question requires clearer motivation and justification. This paper posits that resource-constrained end devices are anticipated to perform real-time inference on dynamically changing task streams. It would be beneficial to elaborate on the specific application scenarios where end devices encounter such time-varying task streams, highlighting the practical relevance and necessity of the research.
2. In the introduction, the authors claims that "Despite promising, current online CL has primarily focused on optimizing the learning performance, overlooking the requirements of system performance, such as the inference latency and resource efficiency". This conclusion is unfair, since many recent works have addressed the tradeoff among learning performance, inference performance and resource constraints, e.g., RECL [22], Cost-effective On-device Continual Learning over Memory Hierarchy with Miro [Mobicom 23], Online Resource Allocation for Edge Intelligence with Colocated Model Retraining and Inference [INFOCOM 24], Usas: A Sustainable Continuous-Learning´ Framework for Edge Servers [HPCA 24], DACAPO: Accelerating Continuous Learning in Autonomous Systems for Video Analytics [ISCA24].

**Reviewer Confidence:**

3: The reviewer is confident but not certain that the evaluation is correct

**Scope:**

4: The work is relevant to the Web and to the track, and is of broad interest to the community

---

### Official Review · Reviewer_8sQq · 2024-12-04

**Novelty:** 2
**Technical Quality:** 3

**Review:**

This paper presents ELITE, a continual learning framework targeting edge scenarios. It proposes the device-cloud collaboration paradigm to address model adaptation and low-latency inference issues on edge devices where resources are constrained. To that end, the framework first adopts multi-task learning to enhance the capabilities of each model in the model zoo on the server. To enable model selection, each client runs a model to extract features of a sample and send them to the server for selecting the most proper model. It also selectively picks layers for fine-tuning to achieve low latency. Experimental results show that the framework can achieve higher accuracy and lower latency compared to existing device-only and cloud-only approaches.

Strengths:
S1. Device-cloud collaboration is an effective strategy to address the resource constraints of edge devices to apply machine learning.

S2. A real hardware setup is adopted in the evaluation, which increases the fidelity of the results.

S3. ELITE is shown to achieve higher performance in terms of both accuracy and latency compared to the selected baselines.

Weaknesses:
W1. The system model and application scenarios of the proposed approach aren't clear. First, the framework clusters "the entire data samples on the cloud server into n data clusters" and uses the clusters to train a zoo of models in the cloud. How can the server generate this data?  Do clients stream their data to the cloud to bootstrap the zoo? If so, the overhead of uploading the streams is unrealistically high. In addition, it's unclear what applications can benefit from the proposal. The workloads involved in the evaluation (image classification and video analytics) are general ML workloads. It's unclear why continual learning is beneficial.

W2. Device-cloud collaboration for machine learning training has been widely adopted [1]. Other techniques such as generating a model zoo, model selection, and fine-tuning are pretty existing techniques. Hence, the novelty of this work is rather limited.

W3. The evaluation can be generally improved.
- It's unclear why Forgetting Rate, one of the three evaluation metrics, is important.
- The network configuration between the device and the server is not specified. It's unclear how much network communication is incurred during the model selection.
- Critical design decisions have not been evaluated. Specifically, what's the impact of information entropy for maximizing diversity? How much time is spent in each step of the low-latency inference on the device (e.g., prediction confidence calculation, feature extraction, as well as fetching a new model from the server)? What's the benefit of selectively fine-tuning model layers?
- The final model accuracy is extremely low (even though it's better than that of baselines). E.g., the accuracy of image classification on CIFAR10 can be higher than 90% on the server, but it's only about 40% in ELITE. This significant performance degradation questions the utility of continual learning on edges.


[1] Edge-to-Cloud Collaboration in Machine Learning. IJIRT 2024.

**Questions:**

Please respond to the weak points above.

**Reviewer Confidence:**

3: The reviewer is confident but not certain that the evaluation is correct

**Scope:**

3: The work is somewhat relevant to the Web and to the track, and is of narrow interest to a sub-community